# Hypoxia and ERα Transcriptional Crosstalk Is Associated with Endocrine Resistance in Breast Cancer

**DOI:** 10.3390/cancers14194934

**Published:** 2022-10-08

**Authors:** Charly Jehanno, Pascale Le Goff, Denis Habauzit, Yann Le Page, Sylvain Lecomte, Estelle Lecluze, Frédéric Percevault, Stéphane Avner, Raphaël Métivier, Denis Michel, Gilles Flouriot

**Affiliations:** 1Inserm, EHESP, Irset (Institut de Recherche en Santé, Environnement et Travail)-UMR S1085, Rennes University, 35000 Rennes, France; 2French Agency for Food, Environmental and Occupational Health & Safety (ANSES), Toxicology of Contaminants Unit, 35300 Fougères, France; 3Institut de Génétique et Développement de Rennes, UMR 6290 CNRS, Université de Rennes, 35000 Rennes, France

**Keywords:** breast cancer, estrogen receptor alpha, endocrine resistance, hypoxia

## Abstract

**Simple Summary:**

Hormone receptor positive breast cancer patients are treated with anti-hormone molecules as a standard of care. However, resistance frequently occurs, leading to hormone resistant metastatic relapses in foreign organs. Understanding the molecular mechanisms through which breast cancer cells evade therapeutic pressure is of paramount interest. Hypoxia, which refers to oxygen deprivation and is characterized by the activation of hypoxia inducible factors, is a common feature of the solid tumor microenvironment, yet its influence on estrogen receptor alpha activity remains elusive. Here, we investigate the consequence of hypoxia and the signaling of hypoxia inducible factors on hormone responsiveness in breast cancer cells and its clinical implications.

**Abstract:**

Estrogen receptor-alpha (ERα) is the driving transcription factor in 70% of breast cancers and its activity is associated with hormone dependent tumor cell proliferation and survival. Given the recurrence of hormone resistant relapses, understanding the etiological factors fueling resistance is of major clinical interest. Hypoxia, a frequent feature of the solid tumor microenvironment, has been described to promote endocrine resistance by triggering ERα down-regulation in both in vitro and in vivo models. Yet, the consequences of hypoxia on ERα genomic activity remain largely elusive. In the present study, transcriptomic analysis shows that hypoxia regulates a fraction of ERα target genes, underlying an important regulatory overlap between hypoxic and estrogenic signaling. This gene expression reprogramming is associated with a massive reorganization of ERα cistrome, highlighted by a massive loss of ERα binding sites. Profiling of enhancer acetylation revealed a hormone independent enhancer activation at the vicinity of genes harboring hypoxia inducible factor (HIFα) binding sites, the major transcription factors governing hypoxic adaptation. This activation counterbalances the loss of ERα and sustains hormone-independent gene expression. We describe hypoxia in luminal ERα (+) breast cancer as a key factor interfering with endocrine therapies, associated with poor clinical prognosis in breast cancer patients.

## 1. Introduction

ERα has a central role in the development of hormone-dependent breast cancer, mainly by triggering the activation of transcriptional programs promoting cell proliferation, resistance to cell death or angiogenesis upon E2 stimulation [1]. ERα-expressing luminal A and B breast tumor subtypes account for 70% of total breast cancers [2], and are treated by hormone therapy aiming either at selectively blocking ERα activity (Selective Estrogen Receptor Modulator—SERM), dampening ERα expression (Selective Estrogen Receptor Degrader—SERD) or suppressing production of endogenous E2 (Aromatase Inhibitor—AI, LH-RH agonists) [3]. It is estimated that approximately 30 to 40% of these tumors initially fail to respond to endocrine therapy because of intrinsic mechanisms of resistance and/or relapse after several months/years of treatment by de novo acquisition of resistance [3]. Activating mutations in the *ESR1* gene (Y537S, D538G) [4], oncogenic activation of growth signaling pathways [5,6], activation of EMT (epithelial-to-mesenchymal transition) programs by CAFs derived cytokines or growth factors [7,8] or via MRTFA translocation [9] are examples of mechanisms contributing to hormone resistance and therapy failure. Of note, very little is known about the contribution of the tumor microenvironment (TME) to anti-hormone therapy resistance. Among the different features of the TME is oxygen deprivation, also known as hypoxia, observed in more than 50% of the solid breast tumors [10]. In normal breast tissue, oxygen tension is estimated to be around 60 to 70 mmHg of pO2 (8/9%) whereas it can drop down to < 8 mmHg of pO2 (<1%) in the case of severe hypoxia [11,12]. Hypoxia has been described to promote a wide range of malignant traits such as the epithelial–mesenchymal transition (EMT), angiogenesis or metabolic reprogramming, classified as “Hallmarks of cancer” [13,14]. These programs are governed by the evolutionarily conserved hypoxia-inducible factors HIF1α and HIF2α (hypoxia inducible factor) [15]. In normoxic conditions, HIFα factors are continuously addressed for proteasomal degradation by oxygen-dependent prolyl-hydroxylase enzymes (PHD) and E3 ubiquitin–ligase von-Hippel Lindau (pVHL) [16]. When oxygen becomes scarce, HIFα factors are stabilized and accumulate in the nucleus where they activate transcription programs to adapt to hypoxia [17].

Hypoxia has been shown to inhibit ERα expression both in vitro and in vivo [18,19,20], yet conflicting studies co-exist in the literature in regard to its consequences on estrogenic signaling. Our work confirms hypoxia-induced down-regulation of ERα using both gas hypoxia and Cobalt Chloride (CoCl2) hypoxia mimetic agents on the ERα-positive breast cancer cell lines MCF7 and T47D, predominantly at the protein level. It has been reported that hypoxia, via the HIFα factors, and ERα share common transcriptional target genes, yet consequences of hypoxia on ERα transcriptional activity and binding to chromatin remain unknown [21,22]. Our data demonstrate that hypoxia differentially regulates a fraction of ERα responsive genes, either by inhibiting or by enhancing their expression. Genome-wide mapping of ERα binding using ChIP-Seq shows important reprogramming of ERα cistrome in hypoxia, with a massive loss of ERα binding sites (ERBSs) and unexpected acquisition of hypoxia-specific ERBS. Targeted H3K27Ac ChIP experiments revealed hypoxia-dependent enhancer activation of ERα target genes, whose expression can predict worse overall survival, and revealed that HIFα factors play a crucial role in the reprogramming of ERα cistrome. In conclusion, our work demonstrates that hypoxia can modulate efficacy of endocrine therapies by interfering with estrogenic signaling.

## 2. Materials and Methods

### 2.1. Cell Culture and Treatments

MCF7 and T47D cells were routinely maintained in DMEM (GIBCO, ThermoFisher Scientific, Waltham, MA, USA) supplemented with 10% fetal calf serum (FCS; Biowest, Nuaillé, France) and antibiotics (GIBCO) at 37 °C in 5% CO2. When treatments with steroids were required, the cells were in phenol red free DMEM (GIBCO) supplemented with 2% dextran/charcoal-stripped FCS (dsFCS; Biowest) prior to the experiments. E2 and 4-hydroxy-tamoxifen (4-OHT) were purchased from Sigma–Aldrich. Proteasomal inhibitor MG-132 was purchased from Enzo Life Sciences France, Villeurbanne, France (BML-PI102-0005). Hypoxic treatments were performed using chemical mimicking agent, cobalt chloride (CoCl2), from Sigma–Aldrich, Saint-Louis, MO, USA. When specified on the figure gas, hypoxia was performed in an H35 hypoxystation from Don Whitely Scientific, Victoria Works, United Kingdom. MCF7 cells were exposed for one month in the presence of 200 µM of CoCl2 to induce hypoxia. Rescued cells were obtained after one month of recovery in CoCl2-free media. siRNA Transfections were carried out using lipofectamine RNAiMAX (Invitrogen, Waltham, MA, USA) reagents according to the manufacturer’s instruction. Human HIFα siRNAs (Mission esiRNA, HIF1α; EHU151981, HIF2α; EHU008751) were obtained from Sigma–Aldrich.

### 2.2. Proliferation Assay and TUNEL Assay

For the proliferation assay, 10,000 MCF7 cells were plated in 24-well plates and then culture in steroid free medium with 2% FBS for 24 h. Cells were then treated with either E2 (10 nM) or 4-OHT (1 µM) for 6 days. After treatment, cells were trypsinized, numeration was performed using a TC10 Automated Cell Counter (Bio-Rad, Hercules, CA, USA). The terminal deoxynucleotidyl transferase mediated d-UTP nick-end labeling (TUNEL) assay was used to measure the level of apoptotic cells as defined by detecting DNA fragmentation (In Situ Cell Death detection kit, Roche, Basle, Switzerland).

### 2.3. Cell Cycle Analysis by Flow Cytometry (FACS)

After 48 h of hormone stimulation, cells were trypsinized and resuspended in IFA buffer (HEPES 10 mM, NaCl 150 mM, FBS 4%). Cells were fixed with ice-cold 70% ethanol for 30 min, rinsed with PBS and permeabilized with IFA buffer/0.5% Tween. RNAs were eliminated by RNase A (100 ug/mL) treatment for 1 h at room temperature, and then cells were incubated with propidium iodide 1 mg/mL, for 30 min. The suspension was filtered through a nylon mesh prior to analysis using a FACS Calibur Flow Cytometer (BD Bioscience). For each condition, 20,000 cells were counted.

### 2.4. Antibodies and Immunofluorescence

The primary antibodies used for Western blotting and immunofluorescence analyses were the following: ERα (HC-20, Santa Cruz, Santa Cruz, CA, USA, sc-543), ERK (Santa Cruz, sc94), HIF1α (Abcam, Cambridge, United Kingdom, ab2185) and HIF2α (Abcam, ab199), pAKT_Ser473 (Cell Signaling, 4060). For western blotting, the peroxidase-conjugated secondary antibodies directed against rabbit or mouse constant fragments were purchased from GE HealthCare, Chicago, IL, USA. For immunofluorescence, secondary antibodies coupled with 594 or 488 Alexa Fluor dyes were obtained from Abcam. The slides were mounted in mounting medium with DAPI (Sigma–Aldrich, Saint-Louis, MI, USA). Images were obtained with an ApoTome Axio Z1 Imager microscope (Zeiss, Göttingen, Germany) and processed with Axio Vision Software (4;8;2 SP1, Zeiss, Göttingen, Germany). Fluorescence was quantified with ImageJ software from images obtained with identical exposure time.

### 2.5. Western Blotting

Cells were lysed in NP-40 lysis buffer (50 mM Tris-HCl (pH 7.5), 150 mM NaCl, 1% Nonidet P40, 0.5% sodium deoxycholate and 0.1% SDS) containing a cocktail of protease and phosphatase inhibitors (Roche), or directly lysed into 2X Laemmli buffer. Whole protein extractions were denaturized at 95 °C for 5 min, subjected to sonication (Bioruptor, Diagenode, Liège, Belgium) and transferred on nitrocellulose membrane (GE HealthCare) after being separated on SDS polyacrylamide gel. Primary antibodies were incubated in TBS 5% milk supplemented by 0.1% of Tween (Sigma–Aldrich).

### 2.6. RNA Extraction and RT qPCR

RNA extractions were performed using RNeasy kit (Qiagen, Hilden, Germany) according to the manufacturer’s instructions. Reverse transcription was performed on 1 µg of RNA with a random primer using iScript Reverse Transcription Supermix purchased from BioRad. Quantitative RT-qPCR were performed using the iQ™ SYBR_ Green Supermix from BioRad using CFX384 Touch Real-Time PCR Detection machine (Bio-Rad, Hercules, CA, USA). TBP was used as internal control, and the primer sequences used for qPCR reaction are indicated in Appendix A.

### 2.7. Microarray Preparation

MCF7 cells were cultured in phenol red free DMEM (GIBCO) supplemented with 2% dextran/charcoal-stripped FCS (dsFBS; Biowest) 72 h prior to the experiments. Then, cells were treated for 8 h with EtOH or 10 nM of E2. After treatment, the control, CoCl2-treated and rescued cells were immediately harvested. Total RNA was purified using the RNeasy kit (Qiagen) and quantified using the Nanodrop 1000 spectrophotometer (Nanodrop Technology, Cambridge, UK), as previously described [23]. The RNA quality was controlled with Bioanalyzer and RNA integrity Number (RIN) were between 8.6 and 9.7. A one-color whole gene expression modification analysis was performed using the Agilent Whole Human Genome 8 × 60 K Microarray kit (Agilent Technologies, Santa Clara, CA, USA). Four replicates per condition were analyzed with the GeneSpring GX software (Agilent Technologies). Briefly, the expression profile was log2-transformed and normalized (Quantile), and 23 531 gene entities were eventually detected on the microarrays. Expression level-based filters (intensity greater than 125) were applied, and finally, the considered list contained 17 291 gene entities. The complete data set was deposited in the Gene Expression Omnibus (GEO) database (www.ncbi.nlm.nih.gov/geo, accessed on 9 October 2022, GEO series accession number: GSE111914).

### 2.8. Microarray Analysis

A gene entity list was filtered by fold change >1.5 between all conditions. Gene expression modifications were then compared using an ANOVA followed by a post-hoc Tukey test. *p*-values were adjusted by controlling the false discovery rate (FDR) with the Benjamini and Hochberg (BH) correction for multiple testing. A total of 15 direct side-by-side comparisons were performed. A gene was considered significantly differentially expressed if the adjusted *p*-value was below 0.05, and the absolute fold-change (FC) was above 1.8 at least in one comparison.

### 2.9. Chromatin Immunoprecipitation (ChIP)

The sub-confluent control, CoCl2-treated and rescued MCF7 cells were grown for 72 h in phenol red-free DMEM (Invitrogen) containing 2% dextran/charcoal-stripped FCS (Biowest), and then treated with 10 nM E2 or ethanol (vehicle control) for 50 min. Cells were then washed twice with ice-cold PBS and cross-linked for 10 min with 1.5% formaldehyde (Sigma) and the reaction was stopped with 0.125 M glycine for 1 min. Following two washes in cold-PBS, cells were scraped into 500 µL PBS with protease inhibitors (Complete Inhibitors, Roche), spun 2 min at 3000 g and snap frozen to −80 °C. After cell lysis in 300 μL of lysis buffer (10 mM EDTA, 50 mM Tris-HCl (pH 8.0), 1% SDS, 0.5% Empigen BB), ChIP was performed as previously described [24]. Antibodies against ERα (HC-20, sc-543, Santa Cruz), H3K27ac (ab4729, Abcam) and H3K4me2 (07-030, Merck Millipore, Burlington, MA, USA) were used in these assays. DNA was purified on NucleoSpin columns (Macherey-nagel, Hoerdt, France) using NTB buffer. ChIP experiments were performed from at least five biological independent replicates. Primers for real-time PCR are provided in Appendix A.

### 2.10. ChIP-Sequencing (ChIP-Seq) and Data Analysis

We pooled DNA originated from at least 20 different ChIP experiments conducted as described above. The ChIP DNA was prepared into multiplexed libraries and sequenced using an Illumina HiSeq apparatus at the GenomEast platform (Institut de Génétique et de Biologie Moléculaire et cellulaire; Strasbourg, France). Pooled control input DNA were processed in parallel. Reads were aligned onto the indexed chromosomes of the human hg19 (GRCh37) genome using bowtie 0.12.7 [25], allowing at most two mismatches (parameters -n 2; -l 28; -m 1 with -best and -strata options). Sequencing statistics are given in Appendix A. Extracted reads were then converted to .wig signal files, using samtools [26] and MACS [27] with default parameters. To restrict the inter-sample biases due to slightly different sequencing depths, the raw signals were normalized to the maximum depth obtained. The normalizing coefficient is indicated within Appendix A. Peak calling was then operated as previously [24] using the algorithm described in [28] at different *p*-value thresholds and the input control file as reference. The .bed files containing the identified genomic coordinates of ERBSs were subsequently filtered against the lists of repetitive sequences obtained from the UCSC (blacklist; http://genome.ucsc.edu/cgi-bin, accessed on 25 April 2016). Regions exhibiting high background signal generated by a poor normalization to input due to excessive sequence overrepresentation (consecutive of the highly rearranged genome of MCF7 cells) were also removed. Motifs analyses were performed using the SeqPos algorithm (http://cistrome.org/ap/, accessed on 28 September 2018) [29], and functional annotations were carried out under the GREAT web-platform (http://great.stanford.edu/, accessed on 15 October 2018) [30]. All other integrative analyses of the ChIP-seq data were performed using home-made scripts and algorithms from the cistrome web-platform (http://cistrome.org/ap/, accessed on 28 September 2018) (Appendix A).

### 2.11. Clinical Analysis

Clinical data concerning the METABRIC and TCGA cohorts were extracted from cBioportal (https://www.cbioportal.org/, accessed on 10 June 2021). Kaplan–Meier Plots for HIF1α and HIF2α were generated from the Breast Cancer Kaplan–Meier Plotter (https://kmplot.com/analysis/, accessed on 20 July 2021).

### 2.12. Statistical Analysis

Statistical analyses were performed using Prism software (Prism 9.4.1, San Diego, CA, USA). Statistical significance was determined using the Mann–Whitney *t*-test and was indicated as follows: (*) *p* < 0.05; (**) *p* < 0.01; (***) *p* < 0.001.

## 3. Results

### 3.1. Hypoxia Dampens ERα Expression Level and Modulates E2 Sensitivity at the Cellular Level

Previous works showed that hypoxia induces ERα loss of expression [18,19,20]. We confirm this decrease in the luminal breast cancer cell line MCF7 exposed 24 h either to Cobalt Chloride (CoCl2) or 1% gas hypoxia using H35 hypoxystation from Don Whitely Scientific (Appendix A). CoCl2 was then used as a hypoxia mimicking agent for the study. ERα down-regulation by CoCl2 was also confirmed in the T47D breast cancer cell line (Appendix A). Long-term exposure of MCF7 to CoCl2 (1 month) led to a similar drop of ERα expression, compared to a short 24 h exposure (Figure 1A and Appendix A) suggesting that ERα degradation reaches a plateau under constant hypoxic pressure. Nuclear translocation of hypoxia-inducible factors HIF1α and HIF2α was confirmed by immunofluorescence for 24 h and 1-month CoCl2 exposure (Appendix A). Given that tumors in vivo are exposed to chronic hypoxia, we privileged long-term hypoxia exposure (1 month) in the following experiments. qPCR analysis of ERα mRNA level revealed a modest decrease upon hypoxic pressure, suggesting that expression loss occurs mainly at the protein level (Appendix A). Treatment of MCF7 cells with the proteasome inhibitor MG-132 rescued hypoxia-induced ERα drop, confirming this observation (Figure 1A and Appendix A). We also demonstrate that ERα loss is reversible as cessation of hypoxic pressure restored its expression (Appendix A). Cell cycle analysis using FACS revealed a loss of S-phase entry upon E2 stimulation under hypoxic pressure, as compared to a 2 to 3-fold increase in normoxia (Figure 1B). This result indicates that CoCl2-induced hypoxia abrogates E2-mediated proliferation. Intriguingly, a 2-fold increase in the percentage of cells in S-phase in CoCl2 treated MCF7 cells in the absence of E2 ligand was observed as compared to the control, indicating that hypoxia can induce proliferation independently of E2 (Figure 1B). Of note, while 4-OHT treatment abrogates S-phase entry in normoxic conditions, it failed to prevent hypoxia-dependent S-phase entry, thus indicating the acquisition of anti-hormone resistance. To confirm whether S-phase entry was translated into increased cell proliferation, we performed cells counting experiments. We first confirmed the loss the of E2-induced proliferation under hypoxia, as compared to normoxia (Figure 1C and Appendix A). However, we could not observe increased cell proliferation in absence of E2 in CoCl2 treated cells, highlighting a loss of hormonal control (Figure 1C). To explain this result, we finally performed terminal deoxynucleotidyl transferase mediated d-UTP nick-end labeling (TUNEL) assessing apoptosis. Strikingly, we observed a 3-fold increase in cell death in hypoxia as compared to the control and a complete loss of E2-mediated protective effects in CoCl2-treated cells (Figure 1D). Corroborating this observation, the abundance of *p*-AKT (Ser473) was reduced in MCF7 cells exposed to CoCl2 (Figure 1A). We also demonstrate that E2-dependent proliferation is restored 1 month after CoCl2 removal (Appendix A). Altogether, these results show that hypoxia can induce S-phase entry in MCF7 cells, independently of the presence of E2, but that it does not translate into increased proliferation, as it is counter balanced by increased hypoxia-induced apoptosis.

### 3.2. Hypoxia-Induced E2-Dependant Transcription Reprogramming

To further elucidate the effects of hypoxia on estrogenic signaling, we assessed the transcriptional response to E2 in MCF7 cells in normoxic and CoCl2-treated MCF7 cells using a microarray. Control and CoCl2-exposed MCF7 were treated for 8 h with 10 nM of E2 or EtOH. Differential gene expression (DE) analysis (FDR 0.05—FC 1.8) and supervised clustering revealed a first category of genes whose expression depends on hypoxia exclusively.

Functional annotation identified classical hypoxic pathways such as metabolic reprogramming and glycolysis (Appendix A). Transcriptional network reconstruction using AMEN [31] further confirmed that a substantial number of those genes are direct or indirect HIF1α and/or HIF2α target genes (Appendix A). The analysis of the transcriptome revealed a second category comprising 584 genes differentially expressed genes upon E2 treatment, 518 in normoxic cells and 311 in CoCl2-treated MCF7 cells. Strikingly, more than half of these genes (273) display a loss of E2-dependent regulation in CoCl2-treated cells (Figure 2A). Hypoxia therefore exerts an overall inhibitory effect on the hormone-dependent regulation of ERα target genes, consistent with the loss of ERα expression and cellular response to E2. Intriguingly, we detected 66 (47 up-regulated, 19 down-regulated genes) hypoxia-specific E2-regulated genes, indicating a partial reprogramming of E2 transcriptional response in hypoxia. Interestingly, scatter plot analysis revealed a similar Log2 fold change for the remaining E2-dependant genes both in normoxia and hypoxia, indicating that ERα transactivation (capacity to regulate a target gene) is not inhibited by hypoxia (Figure 2B). We also demonstrate that E2-mediated transcription is restored 1 month after CoCl2 removal (Appendix A), indicating that hypoxia does not imprint a long lasting memory effect at the transcriptional level.

Among the 584 hormone responsive genes, clustering analysis allowed the identification of six gene clusters, either down-regulated (C1 to C3) or up-regulated (C4 to C6) upon E2 treatment, with subgroups referring to as genes whose basal expression level is down-regulated by hypoxia (C1 and C4), unchanged (C2 and C5) or up-regulated by hypoxia (C3 and C6), respectively (Figure 2C). For each cluster, the expression pattern has been validated using RT-qPCR on a panel of representative genes (Figure 2D). Changes in the expression profile were also confirmed on some selected genes in the T47D cell line (Appendix A). Among the genes of clusters C1 and C4 whose expression is inhibited by hypoxia, we found the progesterone receptor (PGR) or the chemokine CXCL12, which are typical markers of hormone responsiveness. Interestingly, the clusters C3 and C6 corresponding to genes whose expression is up-regulated by hypoxia, indicating that hypoxia can sustain their expression level despite loss of ERα expression. This cluster especially includes the epidermal growth factor AREG and the epigenetic regulator KDM4B. Altogether, these results reveal an important overlap between estrogenic and hypoxic signaling.

### 3.3. Hypoxia-Induced Chromatin-ERα Interaction Reprogramming

To further elucidate the mechanisms accounting for hypoxia-mediated changes in ERα-dependent gene expression and to assess whether the receptor binding to chromatin was modulated by hypoxia, we performed ChIP-Seq targeting ERα in MCF7 control and CoCl2-exposed cells, treated with E2 or EtOH for 50 min [32]. Bioinformatics treatment of the data, including filtration and identification of the optimal *p*-value per conditions (Appendix A), was performed as described in Material and Methods. Comparison of the identified estrogen receptor binding sites (ERBS) in untreated and E2-treated control and hypoxic cells is illustrated by the Venn diagrams in Figure 3A,B. Mean ChIP-seq signal measured on each specific subset of ERBS is shown on the bottom of each Venn diagram. In control MCF7 cells, 3896 and 22,452 ERBS were detected in untreated and E2-treated cells, respectively. In CoCl2-treated MCF7 cells, an important drop in the number of ERBS was observed, 3609 ERBSs were still detected in absence of hormone, while E2 stimulation mobilizes 5716 ERBSs (4-fold drop). ChIP-Seq analysis further shows a partial reprogramming of ERα cistrome upon CoCl2-induced hypoxic stress, as demonstrated by the identification of 1042 new ERBSs in hypoxia (Figure 3B). DNA motif analysis of these new ERBSs show depletion in ERE, while ERE remains the main motif in conserved ERBSs, indicating altered ERα binding (Appendix A). Furthermore, a comparison of the new ERBSs with published binding sites of other transcription factors in MCF7 cells reveals weak clustering, with a notable depletion for pioneer factors GATA3 or FOXA1 binding, indicating that the new ERBSs are particular and non-conventional (Appendix A).

As previously explained, transcriptomic analysis revealed six clusters of differentially expressed genes upon E2 treatment in hypoxia, with clusters C1 to C3 being down-regulated and clusters C4 to C6 being up-regulated. Baseline expression level (in absence of E2) is reduced (C1 and C4), unchanged (C2 and C5) or enhanced (C3 and C6) in hypoxia. To determine whether changes in the baseline gene expression level were correlated with changes in ERα binding, we assessed ERα binding at the vicinity of their transcription start site (TSS) (Figure 3C). Results first show that all the clusters display a loss of ERBSs, especially in C1 to C3. On the contrary, the highest percentage of preserved (common) ERBSs was observed in clusters with genes positively regulated by E2, notably in cluster 6, suggesting that hypoxia differentially modulates ERα binding across the different clusters. Finally, hypoxia-specific ERBSs were specifically enriched at the vicinity of both ERα and hypoxia-dependent genes (C1, C3, C4 and C6) (Figure 3C), corroborating that chromatin accessibility is modified by hypoxia, specifically. This observation shows that hypoxia induces chromatin remodeling, rendering new sites accessible to ERα binding, potentially contributing to the control expression of specific ERα target genes. Quantification of the ChIP-seq signal in clusters C4 to C6 confirms that hypoxia inhibits the dynamic of ERα binding following E2 stimulation, regardless of the considered cluster (Figure 3D). Target ERα CHIP experiments targeting ERBS at the vicinity of genes belonging to clusters C4 (*HCK*, *PGR*, *CXCL12*), C5 (*CDC25a*, *NTN1*, *SGK1*) and C6 (*TFF1*, *AREG*, *FOXC1*, *KDM4B*, *GREB1*) showed decreased ERα recruitment to chromatin in the absence and presence of E2 in CoCl2 treated MCF7 cells as compared to normoxia (Figure 3E). Additionally, we also validated new hypoxia-specific ERα binding sites, validating a reprogramming in ERα cistrome (Figure 3F). We also demonstrate that ERα cistrome (both in absence and presence of E2) is restored 1 month after CoCl2 removal (Appendix A), as the majority of the ERBSs lost in CoCl2-treated cells is remobilized in the rescued cells, underlying that hypoxia does not imprint a long lasting memory effect at the chromatin level. Altogether, these results indicate that hypoxia inhibits genome-wide ERα binding overall, which could account for the partial loss of E2-responsivness at the transcriptomic level, in a reversible manner.

### 3.4. HIFα Factors Compensate ERα Loss and Sustain Expression of Specific ERα Target Genes

We then performed ChIP targeting H3K27Ac, a histone mark delineating active enhancers, and assessed its enrichment at the vicinity of genes belonging to the different transcriptomic clusters C4, C5 and C6 (Figure 4A). Interestingly, H3K27Ac was found to be specifically enriched at ERBS located nearby genes of cluster C6 (FOXC1, AREG, KDM4B), whose expression is up-regulated both by hypoxia and E2. H3K27Ac CHIP targeting genes of clusters C4 and C5 did not show any hypoxia-dependent regulation of those ERBS (Figure 4A). Together, these results indicate that some ERα bound enhancers can be primed by hypoxia, triggering the differential expression of genes located at the vicinity, independently of hormone stimulation.

We next used the ChEA algorithm that interrogates publicly available ChIP-Seq datasets to identify transcription factor promoter binding. Strikingly, we found HIF1α, a transcriptional master regulator of hypoxia to bind chromatin nearby genes of clusters C3 and C6, specifically (Figure 4B and Appendix A). Next, we used HIF1α and HIF2α ChIP-Seq from Mole et al. [33], and used Genomic Regions Enrichment of Annotations Tool (GREAT) to identify HIF1α- and HIF2α- bound genes in MCF7, and overlapped these two gene sets with our E2 differential gene lists. This analysis revealed over 80 ERα-dependent genes sharing binding with HIFα factors, with 30 genes sharing HIF1α and HIF2α binding. The distribution of these genes across the six different clusters revealed enrichment in the clusters C3 and C6, confirming that HIFα factors are responsible for E2 independent preparation of these enhancers (Figure 4C,D). Finally, we show that knock-down of HIFα factors using siRNA partly inhibits CoCl2-induced down-regulation of the E2-regulated genes from cluster C4 (*CXCL12* and *PGR*) and prevents CoCl2-induced regulation of the basal expression of genes belonging to cluster C6 (*AREG* and *GREB1*) (Figure 4E). This result indicates that HIFα signaling directly interferes with estrogenic signaling at the chromatin level.

### 3.5. HIFα-Hypoxia-ERα Crosstalk Correlates with Poor Survival in ER+ Luminal Breast Cancers

Next, we addressed whether the crosstalk between HIFα signaling and ERα signaling reverberates on anti-hormone response by treating the control and CoCl2 exposed MCF7 cells with 4-HydroxyTamoxifen (4-OHT) and then by checking the expression of some candidate genes through RT-qPCR. As shown in Figure 5A, while 4-OHT treatment clearly inhibits E2-induced changes in the expression level of the studied genes from clusters C4 and C6, it has no impact on hypoxia-induced changes in the basal expression of these genes (Figure 5A). Notably, the epidermal growth factor AREG gene from cluster C6 still has sustained expression despite the presence of 4-OHT. Similar gene expression patterns were obtained in the other luminal breast cancer cell line, T47D (Appendix A). These data suggest that anti-hormone molecules become inefficient to exhaustively dampen the expression of some ERα-target genes upon hypoxia.

Next, we investigated the clinical relevance of the different gene signatures identified from microarray analysis, by using the clinically available Metabric cohort comprising 1980 tumor samples, composed in the majority of ERα-positive invasive ductal carcinoma (Figure 5B) [34]. Interestingly, we found that alterations in the expression (z-score = 2) of genes belonging to clusters C3 and C6 can predict worse overall survival in the Metabric cohort, as compared to the clusters C1 and C4, significantly for the cluster C3 and with a log rank test *p*-value of 0.069 for the cluster C6 (Figure 5C). This observation supports the idea that hypoxia, by up-regulating those E2 dependent gene sets whose expression can no longer be hampered by hormone therapy, worsens patient prognosis. We additionally found that *HIF1a* high expression and not *EPAS1* (HIF2a) predicts overall (OS) and relapse-free (RFS) survival in the Metabric cohort (Figure 5D,E). This observation was confirmed using the Breast Cancer Kaplan–Meier Plotter resource [35] that allows for the discrimination of tumors according to their ERα status (Figure 5F,G). Of note, altered *HIF1a* expression but not *EPAS1* (HIF2a) was also associated with poor prognosis in the TCGA cohort [34] (Appendix A). Finally, we found that *HIF1a* expression was anti-correlated with *ESR1* expression in both TCGA and Metabric cohorts (Appendix A). Together, these clinical data show that HIF1α mediated hypoxic crosstalk with the estrogenic signaling correlates with poor overall survival of patients with in ERα-positive breast cancers.

## 4. Discussion

Therapy resistance remains a clinical challenge and there is a critical need to identify etiological factors accounting for such phenomenon in each of the different breast cancer subtypes [36,37,38]. While different cancer cell-intrinsic mechanisms have been proposed to favor endocrine resistance, ranging from gene amplification [39], parallel growth factor signaling [40,41], phenotypic plasticity [42] to epigenetic [43,44] and metabolic reprogramming [45], very little is known about the contribution of the tumor microenvironment (TME).

Previous work on breast cancer cell lines highlighted a role for hypoxia in inhibiting ERα expression and E2-mediated proliferation [46]. We confirm here that CoCl2-induced hypoxia inhibits ERα expression, mainly at the protein level. Interestingly, loss of ERα expression occurs in approximately 10% of endocrine-resistant breast cancers, largely through unknown mechanisms [47]. Previous studies have suggested that loss of ERα protein expression is not directly associated with HIF1α since stable transfection of HIF1α in MCF7 cells does not inhibit ERα expression in normoxia [22], or it can depend on the HIF1α E3 ubiquitin ligase pVHL [48]. Identifying such degradation pathway might be key to prevent loss of ERα expression and thus prevent endocrine resistance.

Conflicting studies co-exist regarding the role of hypoxia in inducing breast cancer cell proliferation. Our results show that CoCl2 treated cells have increased S-phase entry compared to control cells, independently of E2 treatment. Analysis of the transcriptome under hypoxia revealed up-regulation of the growth factor amphiregulin (AREG) as well as its cognate receptors EGFR and ErbB4, which may account for this observation. Interestingly, it was reported that HIF2α enhances autocrine growth signaling through AREG and EGFR/ErbB4 expression in breast cancer cell lines [40,49], strengthening this hypothesis. In contrast, other authors reported a hormone-independent proliferation of MCF7 stably expressing HIF1α [22,50], in absence of hypoxia. In our study, we did not observe hypoxia-induced cell proliferation despite increased S-phase entry. Several mechanisms might explain these apparent discrepancies. First, we show that CoCl2-induced hypoxia promotes cell death and abrogates of E2-mediated protection against apoptosis. Additionally, transcriptome analysis revealed that CoCl2-treatment enhances the expression of *BNIP3*, encoding an extracellular ligand positively regulated by HIF1α which can act as a potent apoptosis inducer [51,52]. Moreover, we observed that hypoxia inhibits PI3K pathway signaling, which is consistent with reduced survival [52,53]. Finally, transcriptome analysis revealed up-regulation of cell-cycle arrest genes such as p21 and GADD45a [54,55], and cell growth machinery arrest genes such as DDIT4 [56] in hypoxia. Collectively, these results indicate that S-phase entry in hypoxia can be counterbalanced by multiple mechanisms. These observations are consistent with previous studies suggesting that epithelial cells are more likely to reach a quiescent state when placed in hypoxia, whereas transformed cells tend to proliferate more intensively [57].

Our data highlighted an important overlap between estrogenic and hypoxic signaling at the transcriptomic level. First, by inhibiting ERα protein abundance, hypoxia exerts an overall inhibitory effect on E2 sensitivity, as more than half of the E2-differentially regulated genes are lost in CoCl2-treated cells. Second, hypoxia, through HIF1/2α factors, can up- or down-regulate an important fraction of ERα target genes, which has major clinical implications, as the use of SERMs such as 4-OHT become inefficient to block their expression (clusters 3 and 6). Interestingly in those clusters, we found some genes described to promote malignant tumor progression, such as FOXC1, which is considered as a main EMT inducer [58], KDM4B, an also well-known oncogene promoting dedifferentiation [59,60] or AREG, a growth factor capable of promoting hormone-independent proliferation [40]. We also found that PGF, a potent angiogenesis-promoting factor [61] is regulated both by ERα and hypoxia. Likewise, VEGFa was previously reported to be regulated both by ERα and HIF1α [62]. Moreover, Yang et al., et al. previously identified a subset of genes insensitive to 4-OHT treatment under hypoxic conditions, thus corroborating our results [22]. Interestingly, we observed that alterations in the expression of genes belonging to clusters 3 and 6, whose baseline expression level is up-regulated by hypoxia (clusters 3 and 6) predict worst overall survival in the Metabric cohort. The question whether genes belonging to one of these clusters could be used a predictive marker for anti-hormone response therefore remains open. Finally, we observed that high expression of HIF1α, but not HIF2α, predicts worse overall survival in ERα positive patients, (Metabric, TCGA and Kaplan–Meier Plotter datasets). These data confirm that when solid breast tumor become hypoxic, patient overall survival decreases. Clinical trials assessing the clinical benefit of HIFα inhibition are currently ongoing in cancer types other than breast, such as glioblastoma or kidney cancer [63]. Further preclinical studies aiming at assessing the potential of HIFα inhibitors in vivo in advanced ERα (+) metastatic endocrine resistant model are necessary to establish the clinical benefit of such intervention.

Finally, our data suggest that the overlap between estrogenic and hypoxic signaling originates at the chromatin level. Interestingly, by performing H3K27Ac ChIP experiments, we found that hypoxia can prime enhancers of ERα target genes (cluster C3 and C6), independently of E2, thus explaining the inability of 4-OHT to block their expression. Further investigations are required to assess whether H3K27 acetyltransferase inhibitors would synergize with hormone therapy to prevent relapses; however, this is beyond the scope of this study. Interestingly, an overlap between HIF1α/HIF2α binding sites and ERα binding sites was already described at the vicinity of more than 200 genes in MCF7 cells [22]. Our data also revealed a partial reprogramming of ERα cistrome, a phenomenon that has been already been observed in endocrine resistant tumors and in distant ERα metastases [64,65]. Such epigenetic plasticity demonstrates that genome-wide ERα binding is associated to the cellular context, and can be programmed by external cues.

## 5. Conclusions

In conclusion, our study shows that hypoxia has a profound impact on the E2 genomic response in breast cancer cells. Hypoxia induces ERα down-regulation which inhibits its genomic activity and binds to chromatin, and can modulate the efficacy of endocrine therapies by selectively sustaining the expression of a subset of ERα responsive genes whose expression is associated with a poor overall survival of patients with ERα (+) breast cancers.

## Figures and Tables

**Figure 1 cancers-14-04934-f001:**
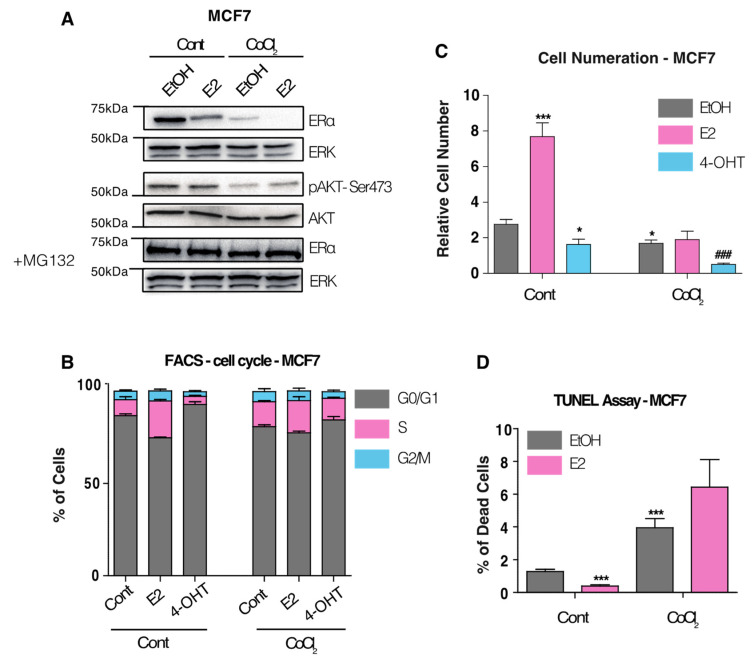
Effect of hypoxia on ERα expression, proliferation and survival of MCF7 cells. (**A**) Western blot showing ERα protein abundance after 1-month exposure to hypoxia inducer CoCl2, in the presence or absence of 10 nM of E2 for 24 h. For proteasome inhibition, cells were treated for 8 h with 2.5 µM of MG-132. ERK1/2 was used as the normalizing control. pAKT Serine 473 and total AKT abundance were also measured. (**B**) Proliferation count of MCF7 cells treated for 6 days with EtOH, 10 nM E2 or 1 µM 4OHT, in presence of 2% of FBS steroid-free. Values are expressed in fold change compared to the number of seeded cells at the beginning of the treatment. (**C**) Cell cycle analysis using FACS using propidium iodide. MCF7 cells were treated for 48 h with EtOH, 10 nM E2 or 1 µM 4-OHT. (**D**) TUNEL assay showing percentage of cell death between control and CoCl2-treated cells, in presence or absence of 10 nM of E2 for 72 h. * *p*-value < 0.05 and *** *p*-value < 0.001 with a Mann–Whitney test for comparisons against the control. ### *p*-value < 0.05 with a Mann–Whitney test for comparisons against CoCl2 treatment.

**Figure 2 cancers-14-04934-f002:**
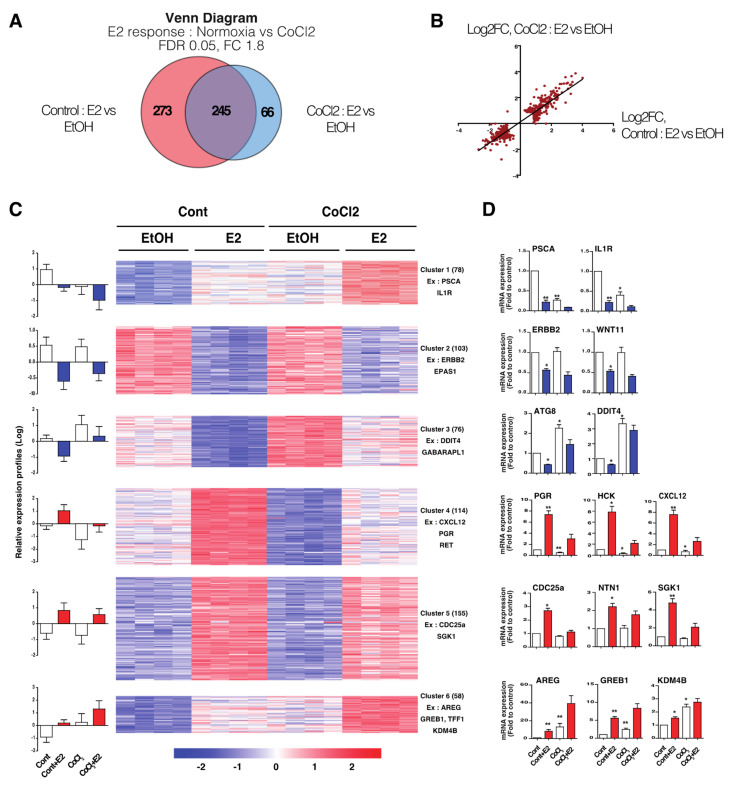
Transcriptomic analysis of the E2 response under hypoxic stress. (**A**) Venn diagram representing E2-regulated genes both in control and CoCl2-treated MCF7 cells (FC > 1.8–FDR < 0.05). (**B**) Scatter plot showing the Log2FC upon E2 treatment for each gene both in control and CoCl2-treated MCF7 cells. (**C**) Heatmap showing supervised clustering (k-means method) of total E2-regulated genes both in control and CoCl2-treated MCF7 cells (FC > 1.8–FDR < 0.05). Control and CoCl2-treated MCF7 cells were treated with EtOH or 10 nM of E2. Transcriptomes from four independent biological samples were analyzed by microarray. Graphs on the left represent the average expression value (Log2) of all genes for each cluster. The number of genes in each cluster are indicated. (**D**) RT-qPCR experiments of representative genes validating the expression profiles found in the six different clusters, * *p*-value < 0.05 and ** *p*-value < 0.01 with a Mann–Whitney test for comparisons against the control. Diagram representing E2-regulated genes both in control and CoCl2-treated MCF7 cells (FC 1.8–FDR 0.05).

**Figure 3 cancers-14-04934-f003:**
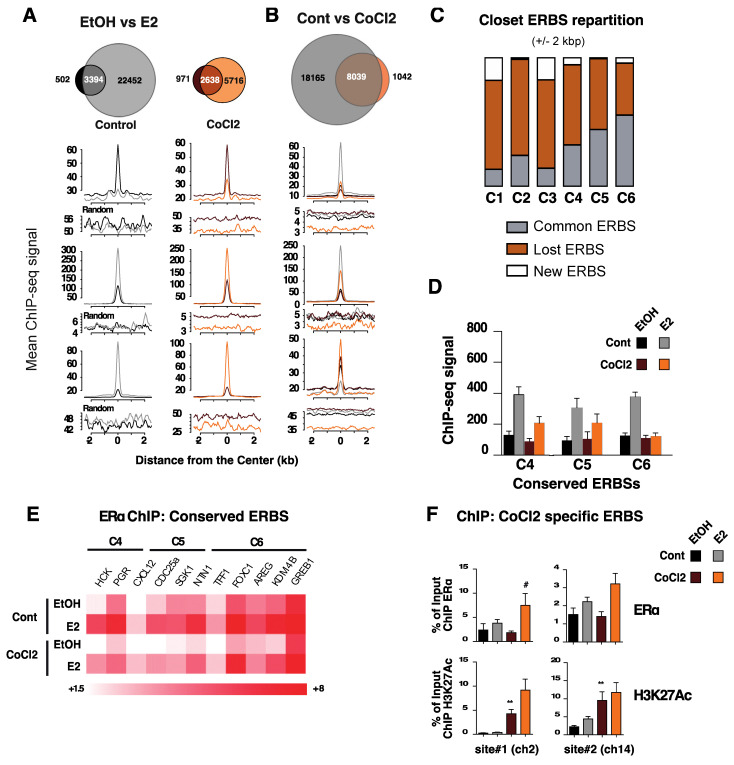
Genome-wide reprogramming of ERα cistrome under hypoxic stress. (**A**,**B**) Venn diagrams showing the overlaps of genomic regions bound by Erα detected by ChIP-seq, both in control or CoCl2-treated MCF7 cells, following a 50 min treatment with E2 or EtOH. The Erα ChIP-seq signal was aligned and averaged within a −2/+2 kbp window centered on ERBS belonging to the 3 categories in the upper Venn diagrams. (**C**) The stacked histogram illustrates the percentage of conserved, lost and gained ERBSs according to each cluster. (**D**) Graphs representing the mean of ERα ChIP-seq signal obtained at the center of the conserved ERBSs located at the vicinity of the TSSs of the genes clustered in C4, C5 and C6 obtained from transcriptomic analysis. (**E**) Heatmap of ERα targeted CHIP-qPCR experiments on conserved ERBSs at the vicinity of genes belonging to the clusters C4, C5 and C6. Data shown are the mean of relative enrichment of five independent experiments normalized to an internal control. (**F**) ERα and H3K27Ac CHIP experiments on CoCl2-specific ERBSs. Data shown are the mean of relative enrichment of five independent experiments normalized to an internal control. ** *p*-value < 0.01 with a Mann–Whitney test for comparisons against the control. # *p*-value < 0.05 with a Mann–Whitney test for comparisons against CoCl2 treatment.

**Figure 4 cancers-14-04934-f004:**
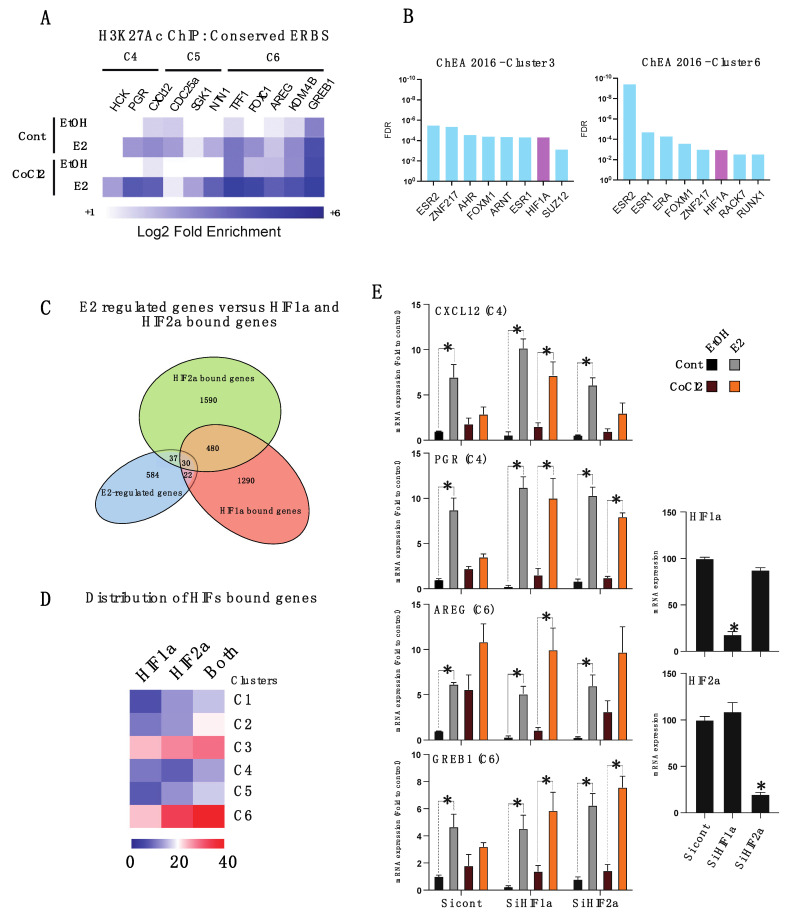
HIF1α regulates a subset of ERα target genes and activates ERα-bound enhancers. (**A**) Heatmap of H3K27Ac targeted CHIP-qPCR experiments on enhancers located at the vicinity of genes belonging the clusters C4, C5 and C6. Data shown are the mean of relative enrichment of five independent experiments normalized to an internal control. (**B**) Analysis of transcription factor binding enrichment using ChEA algorithm on genes belonging to the clusters 3 and 6. (**C**,**D**) Venn diagram showing overlap between the 584 hormone-induced differentially expressed genes and the respective HIF1α and HIF2α bound genes in MCF7 as determined from (Mole et al., [33], using GREAT online tool and their respective distribution across the six different gene clusters. (**E**) Bar graphs showing mRNA expression of ERα representative target genes belonging to clusters 4 and 6, both in control or CoCl2-treated MCF7 cells, treated with control or HIF1α/ HIF2α targeting siRNAs. qPCRs for HIF1α and HIF2α mRNA silencing are shown as the control. Values represent the mean +/− SEM of three experiments and are expressed as fold induction compared to untreated cells (Cont, EtOH) transfected with control Si RNA (* *p* < 0.05, student′s *t*-test).

**Figure 5 cancers-14-04934-f005:**
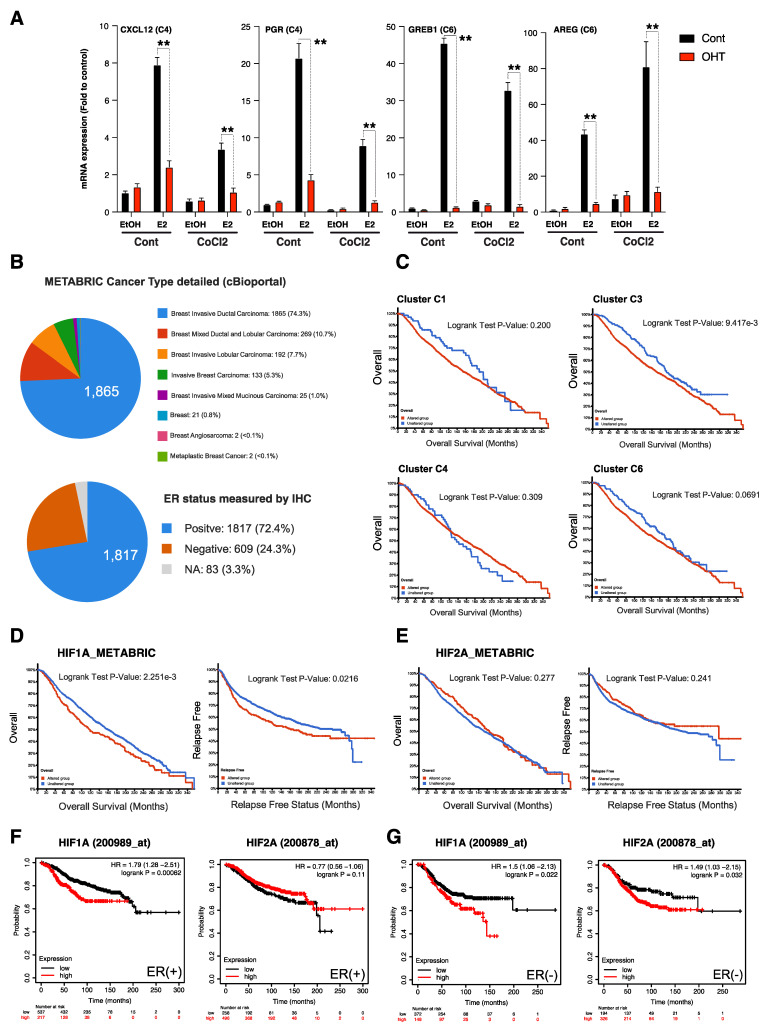
Hypoxia prevents a tamoxifen inhibitory effect on a subset of ERα target genes associated with poor survival prognostic of patients with luminal breast cancers. (**A**) RT-qPCR shows CXCL12, PGR, GREB1 and AREG expression level after 24 h exposure to CoCl2, in presence or absence of 10 nM of E2 and/or 1 μM 4-hydroxytamoxifen (4-OHT) for 24 h. Values represent the mean +/− SD of triplicate and are expressed as a fold induction as compared to untreated cells. (** *p* < 0.05, student’s *t*-test). (**B**) Histopathological features and ERα status measured by immunohistochemistry of the METABRIC cohort (n = 1980 patients) extracted from cBioportal. (**C**) Prognostic values of the different gene signatures (Cluster C1, C3, C4, C6) corresponding to E2 and hypoxia regulated genes identified in Figure 2, in the METABRIC cohort (mRNA expression z-scores relative to diploid samples z-score = 2, n = 1980). (**D**) HIF1α mRNA expression prediction value regarding overall survival (OS) and relapse free survival (RFS) in the METABRIC cohort (mRNA expression z-scores relative to diploid samples, z-score = 1.5, n = 1980). (**E**) HIF2α mRNA expression prediction value regarding overall survival (OS) and relapse free survival (RFS) in the METABRIC cohort (mRNA expression z-scores relative to diploid samples, z-score = 1.5, n = 1980). (**F**) HIF1α and HIF2α mRNA expression prediction value regarding overall survival (OS) in ERα (+) patients from the Breast Cancer Kaplan–Meier Plotter resource (Auto-select cutoff, n = 754). (**G**) HIF1α and HIF2α mRNA expression prediction value regarding overall survival (OS) in ERα (−) patients from the Breast Cancer Kaplan–Meier Plotter resource (Auto-select cutoff, n = 520).

## Data Availability

The microarray and ChIP-Seq data are available on the GEO platform under the following respective numbers: GSE111914 and GSE208663.

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
