# Peer review of "Hypoxia and ERα Transcriptional Crosstalk Is Associated with Endocrine Resistance in Breast Cancer"

_cancers, 2022, doi:10.3390/cancers14194934_

Round 1

Author Response

See file Answer to Reviewer 1

Reviewer 2 Report

In this manuscript, Jehanno et al. conducted transcriptomic analysis and found that hypoxia regulates a fraction of ER targets genes, underlying an important regulatory overlap between hypoxic and estrogenic signaling. Enhancer acetylation analysis revealed a hormone-independent enhancer activation near the hypoxia-inducible factor (HIF1A) binding site, a major transcription factor regulating hypoxia adaptation. This activation counter balances the loss of ER and sustains hormone-independent gene expression. The conception and findings are of enlightening significance to luminal breast cancer therapy, especially for overcoming endocrine resistance. However, the following issues are required for explaining:

1.      The authors should explore whether HIF1A/B is upregulated in endocrine-resistant breast cancer cells by analyzing the GEO public database.

2.      The authors should investigate whether HIF1A/B can predict survival in TCGA database cohort.

3.      Figure 5D/E/F/G: Whether all patients included in the Kaplan-Meier Plotter analysis had received adjuvant endocrine therapy?

4.      Dose-response curve should be depicted to further examine the effect of HIF1A or H3K27Ac inhibition on endocrine resistance in MCF-7 luminal breast cancer cell line.

5.      Can HIF1A/B or H3K27Ac become a druggable target for treating endocrine resistance luminal breast cancer? Are there any inhibitors that can block this target? The authors should discuss more about the future expectations of this target.

6.      Western-blot: The molecular weight of the protein should be labeled.

7.      Some important references about drug resistance in breast cancer should be cited and discussed. For example, PMID: 36124682, 35562334, 36071033, 31969692, 33800852.

8.      Although the manuscript is understandable, there are still some grammatical and spelling errors throughout which could be easily corrected.

Author Response

See File Answer to Reviewer 2
